# Molecular Crosstalk between the Immunological Mechanism of the Tumor Microenvironment and Epithelial–Mesenchymal Transition in Oral Cancer

**DOI:** 10.3390/vaccines10091490

**Published:** 2022-09-07

**Authors:** Kaviyarasi Renu, Sathishkumar Vinayagam, Vishnu Priya Veeraraghavan, Anirban Goutam Mukherjee, Uddesh Ramesh Wanjari, D. S. Prabakaran, Raja Ganesan, Abhijit Dey, Balachandar Vellingiri, Sabariswaran Kandasamy, Gnanasambandan Ramanathan, George Priya Doss C, Alex George, Abilash Valsala Gopalakrishnan

**Affiliations:** 1Centre of Molecular Medicine and Diagnostics (COMManD), Department of Biochemistry, Saveetha Dental College & Hospitals, Saveetha Institute of Medical and Technical Sciences, Saveetha University, Chennai 600077, Tamil Nadu, India; 2Department of Biotechnology, Centre for Postgraduate and Research Studies, Periyar University, Dharmapuri 635205, Tamil Nadu, India; 3Department of Biomedical Sciences, School of Biosciences and Technology, Vellore Institute of Technology (VIT), Vellore 632014, Tamil Nadu, India; 4Department of Radiation Oncology, College of Medicine, Chungbuk National University, Chungdae-ro 1, Seowon-gu, Cheongju 28644, Korea; 5Department of Biotechnology, Ayya Nadar Janaki Ammal College (Autonomous), Srivilliputhur Main Road, Sivakasi 626124, Tamil Nadu, India; 6Institute for Liver and Digestive Diseases, Hallym University, Chuncheon 24252, Korea; 7Department of Life Sciences, Presidency University, Kolkata 700073, West Bengal, India; 8Human Molecular Cytogenetics and Stem Cell Laboratory, Department of Human Genetics and Molecular Biology, Bharathiar University, Coimbatore 641046, Tamil Nadu, India; 9Institute of Energy Research, Jiangsu University, No 301, Xuefu Road, Zhenjiang 212013, China; 10Department of Integrative Biology, School of BioSciences and Technology, Vellore Institute of Technology (VIT), Vellore 632014, Tamil Nadu, India; 11Jubilee Centre for Medical Research, Jubilee Mission Medical College and Research Institute, Thrissur 680005, Kerala, India

**Keywords:** oral cancer, immunological aspects, microenvironment, epithelial-to-mesenchymal transition, signaling events

## Abstract

Oral cancer is a significant non-communicable disease affecting both emergent nations and developed countries. Squamous cell carcinoma of the head and neck represent the eight major familiar cancer types worldwide, accounting for more than 350,000 established cases every year. Oral cancer is one of the most exigent tumors to control and treat. The survival rate of oral cancer is poor due to local invasion along with recurrent lymph node metastasis. The tumor microenvironment contains a different population of cells, such as fibroblasts associated with cancer, immune-infiltrating cells, and other extracellular matrix non-components. Metastasis in a primary site is mainly due to multifaceted progression known as epithelial-to-mesenchymal transition (EMT). For the period of EMT, epithelial cells acquire mesenchymal cell functional and structural characteristics, which lead to cell migration enhancement and promotion of the dissemination of tumor cells. The present review links the tumor microenvironment and the role of EMT in inflammation, transcriptional factors, receptor involvement, microRNA, and other signaling events. It would, in turn, help to better understand the mechanism behind the tumor microenvironment and EMT during oral cancer.

## 1. Introduction

Oral cancer is one of the most important health problems. It is considered the main reason for deaths from oral diseases. Oral squamous cell carcinoma (OSCC) embodies a large number of common malignancies, i.e., 90% of cancer-affected people have OSCC from all the forms of oral cancer. This development of this cancer is mostly caused by increased abuse of alcohol, tobacco use, smoking, and HPV infection [1,2]. In 2018, according to the global statistics for cancer, there were 2% of new cases and 1.9% of cancer. Approximately 90% were OSCC in oral cancer. In one in three patients, oral cancer becomes recurrent and untreatable. This is mainly due to neoplasm recurrence, metastasis of the cancer, and resistance to drugs, which further reduces the survival rate [3]. In recent years, the viewpoint on cancer has changed, and the tumor is not considered as a mass of cancer cells. However, it is a complex tumor environment (TME).

A TME has the component of various cell types. Some of them are cancer with fibroblasts, regulatory Tcells, neutrophils, macrophages, myeloid-derived suppressor cells, platelets, natural killer cells, and mast cells. There is an interaction between these subpopulation cells and other cells, including cancer cells. It occurs through complex network communications via growth factors, chemokines, cytokines, and extracellular matrix (ECM) proteins [4]. Epithelial–mesenchymal transition (EMT) is one of the biological phenomena in which epithelial cells need mesenchymal traits. The epithelium cells categorize cell–cell adhesions tightly and polarity in the apical with the basal. The mesenchymal cells characterize the interaction between the cell–cell in a loose manner, which further causes the augmented action of the migration of cells [5]. It is known that EMT is a dynamic reversible process that mediates epithelial cells to undergo mesenchymal cells via different biochemical changes, increased capacity of migration, apoptotic resistance, invasiveness, and increased ECM production [3].

Although a study shows that EMT is not necessary for the metastasis of the tumor, it is, however, highly involved in cancer metastasis and invasion. Currently, different studies show that the tumor cells involved in the circulation and migration of cells cause tumor metastasis which further leads to EMT [3]. Therefore, we consider EMT as a hotspot in the research of cancer as it is highly involved in metastasis which is the main principle of oncotherapy. The goal of our study is to providean overall view of EMT in oral cancer with different signaling mechanisms, transcriptional factors, etc. This review focuses on the factors mediating TME, the subpopulation of cells with immunological aspects, epithelial–mesenchymal cell transition, and its function in oral cancer development.

## 2. Epithelial–Mesenchymal Transition in Oral Cancer

EMT is a dynamic process that has been found to be reversible. It allows an epithelial cell in its polarized form to undergo different biochemical changes, leading to the phenotype of mesenchymal cells. It includes increased migration capacity, elevated apoptotic resistance, invasiveness, and extra production of ECM [6]. During the invasion stage, epithelial cells with tumors defeat the junction at the intercellular level and polarity between the apical and basal. Therefore, they separate from the membranes at the basement level. With the alteration in the interaction between the cell and ECM, mesenchymal phenotype overexpression is formed through the transcriptional factors of EMT [7]. The cancer cells enter the bloodstream and form the macro- and micrometastases via the bloodstream and endure in the remote site. This is called the enabling of the angiogenic switch. Once the distinct site is reached, there is a development of the metastatic tumor via the growth of cancer cells [8]. Few studies show that EMT is an important condition for tumor metastasis. However, EMT is highly significant in cancer invasion and its metastasis [9]. A recent studies shows that circulating tumor cell formation and cells which collectively migrate play an important role in the contribution to tumor metastasis. This further mediates some of the changes at the phenotypic level along with EMT [3]. At the molecular level, there are three mechanisms involving EMT in oral cancer such as: 1. a decreased level of E-cadherin (the main feature of EMT in oral cancer is the adhesion of cell–cell and elevated motility of cells via E-cadherin downregulation and N-cadherin upregulation); 2. microRNA expression modification (deregulation of the miRNA has been highly connected to the metastasis and resistance of the tumor by changing EMT in oral cancer); 3. actin reorganization and invadopodia formation (this intracellular filament of actin is reorganized via β-catenin and E-cadherin adhesion, and MMPs are involved in forming invadopodia in oral cancer) [5]. Therefore, we say that EMT is an important field in cancer research.

## 3. The Role of Inflammatory Proteins in Influencing EMT in Oral Cancer

### 3.1. Role of Transforming Growth Factor β (TGF-β) in Oral Cancer

TGF-β and its associated pathways play a twin role in the variation of its characteristics at its cellular level, and this is called the TGF-β paradox [10]. The TGF-β pathway is one of the most important pathways which mediates EMT. TGF-β plays an important role in the apoptosis of cancer cells and suppression of tumors in an environment with inflammation. In contrast, TGF-β enhances EMT by encouraging the migration of cancer cells via both Smad signaling and non-Smad signaling pathways [11]. TheSmad and non-Smad signaling pathwaysare initiated by the ligands of the superfamily of TGF-β inclusive of TGF-β (three isoforms) and bone morphogenic protein (BMP-2 to BMP-7; six isoforms) [12]. BMP-2 and BMP-7 are highly related to tumor differentiation and lymph node metastasis during oral cancer, showing a poor diagnosis [13]. Impairing Smad signaling of TGF-β leads to defeating the TGF-β inhibition effects and its proliferation. Either impairment of TGF-β or its attenuation affects its function in regulation since it is involved in the progression of tumor and carcinogenesis. Treatment of TGF-β in OSCC cells manifests a characteristic alteration of EMT by converting cells such as fibroblasts with E-cadherin attenuation and vimentin augmentation [14]. In OSCC, an induction of THBS-2 via TGF-β encourages cancer migration and increases the level of MMPs. This means it favors the invasion of OSCC [13]. The double function of TGF-β is often found in oral cancer metastasis and is associated with its progression [15]. Superfamily members of TGF-β, such as BMP-2, impair the levels of Snail and N-cadherin and attenuates the level of CK9, further showing that BMP-2 is more involved in MET than EMT [16].

### 3.2. The Connection between the Tumor Necrosis Factor—Alpha (TNF-α), Interleukin-1β (IL-1β), Interleukin-6 (IL-6), Interleukin-8 (IL-8), Monocyte Chemoattractant Protein-1(MCP-1/CCL2), Macrophage, and Oral Cancer

Similar to TGF-β, TNF-α plays a twin role in malignant tumors. TNF-α can destroy tumor cells (OSCC) due to its response to the immune system and inflammation. TNF-α generates the invasiveness of cancer and metastasis of cancer through triggering signaling pathways such as MAPK, which induces EMT [17]. In OSCC, TNF-α elevates the mesenchymal marker expression and attenuates the epithelial marker expressions [18]. Some studies show that TNF-α impedes the activity of OSCC migration [19]. TNF-α and TNFR1 play an important role in the progression of OSCC. The neutralization of TNF-α reduces the cytokines in serum, inhibiting invasive lesion progression and further decreasing the neutrophils associated with the tumor in vivo. This shows the role of TNF-α in the transformation of oral malignancy via regulating TNFR1. This acts as a diagnostic feature for OSCC [20].

In the immune cells, IL-1β is triggered by transcription factors such as NF-κB and AP-1. IL-1β is highly expressed in the stromal cells of tumors and effectors cells that infiltrate the tumor immune system. It is highly responsible for tumor microenvironment shaping. In OSCC and oral keratinocytes, IL-1β strengthens EMT via proinflammatory cytokine production such as IL-8, GROα, and IL-6 [21]. The expression of the oral epithelial IL-1β was found to be decreased in increased oral mucosa malignant transformation [22].

IL-6 is one of the multifunctional cytokines which is often found in different tissues and acts in a paracrine or autocrine manner. There is a relationship between the survival of cancer tumors, metastasis of tumor, relapse, OSCC therapeutic resistance, and IL-6 [23]. Though there are different mechanisms of IL-6, the exact mechanism behind cancer progression remains unclear. EMT activation by IL-6 via JAK/STAT3 activation plays an important role in the generation of OSCC [24]. There is an increased IL-6 level in the saliva of OSCC patients compared to normal patients [25].

IL-8 triggers signaling pathways such as JAK/STAT, PI3K and Ras, MAPK, and Raf via binding with GPCR such as CXCR1/2. Erlotinib is a target for EGFR tyrosine kinase, which triggers the IL-8 secretion associated with EMT via the p38 and MAPK kinase pathway. In some cases, such as nasopharyngeal carcinoma, EMT activates via different pathways, such as E-cadherin epigenetic silencing [26]. The expression of the oral epithelial IL-8 was found to be decreased in increased oral mucosa malignant transformation [22].

CCL2 is a cytokine with different types of immune cells which cause acute inflammation. CCL2 binds with the receptors CCR2 and CCR4 to produce a diverse response in the biological system [27]. Along with that, it also binds with other receptors such as ACKR2 and ACKR1. So, CCL2 and its receptor have an important role in inflammation. More evidence shows that CCL2 is associated with cancer metastasis in OSCC. It triggers many signaling pathways to mediate EMT [28]. The increased expression of MCP-1 mediates the progression of OSCC via increasing the signaling pathways involved in pro-survival. The increased expression of MCP-1 leads to a decreased survival rate in oral cancer patients [29].

Macrophages are associated with tumors, generating many mediators that regulate the biological activity of tumors. Tumor-associated macrophages trigger the M1 (classical) or M2 (alternative) polarization phenotype, which each have an activity opposite in nature, such as pro-tumor activity and anti-tumor activity [30]. The tumor-associated macrophage accumulations are closely related to poor outcome in clinical patients. There is an attenuation of ZO-1 and E-cadherin (epithelial markers) and augmentation of the vimentin and N-cadherin (mesenchymal markers) in tumor-associated macrophages in OSCC cells. The transcription factors induced by EMT are Slug and Twist, which are elevated in tissues of OSCC. Macrophages are one of the important inflammatory cells which play an important role in the progression of oral cancer. They are involved in oral cancer development, oral precancerous lesions, etc.; the accumulation of tumor-associated macrophages can cause tumor metastasis due to EMT activation [31] (Table 1).

## 4. EMT-Induced Tumor Hypoxia and Oral Cancer

A hypoxia-related gene plays an important role in diagnosing oral cancer [32]. It shows that hypoxia-related genes play an important role in the enhancement and development of oral cancerincluding its stabilization [33]. Hypoxia-inducing factor-1 (HIF-1) acts as a major regulator for hypoxia in OSCC conditions, even though its downstream effects have not been determined. However, different immunological actions, such as immune surveillance and filtration, have been determined in OSCC as a diagnostic marker for its therapeutic actions [34]. Hypoxia is closely related to elevated immune escape, impairs the immune action in OSCC, and further acts as an anti-oral cancer effect. Suppression of the OSCC anti-tumor effects originate from the tumor itself. It is due to apoptosis or defects in its function in the circulating and Tcells (tumor infiltrating). Along with this tumor microenvironment, with its soluble factor and its hypoxic factors, it causes an agglomeration of cells involved in immune suppression, such as Tregs, MDSCs, macrophages, and TAM. It also downregulates the activity of DCs and T lymphocytes and their function [35].

## 5. Alteration of Cytoskeleton Involved in EMT Acts as a Diagnostic Marker for Oral Cancer Therapy

Cytoskeleton remodeling is a complex process of the metastasis of a tumor. Upon cytoskeleton remodeling, there is a change in the proteins of the skeleton, such as the microfilament, intermediate filament, and microtubule [36]. The specific cytoskeleton proteins involved in epithelial tumor prognosis and oral cancer diagnosis, i.e., OSCC, are vimentin and cytokeratin [37,38]. An intermediate filament protein (type III) is vimentin which is highly expressed in various mesenchymal cells, although not in epithelial cells (normal). There is an anomalous vimentin expression in different tumors, including OSCC. The abnormal expression of vimentin is one of the prognostic factors of OSCC and is determined by multivariate analysis. In OSCC, there is a negative correlation between E-cadherin and vimentin.

Further research shows that the combined expression of vimentin and E-cadherin is utilized for themeasurement of EMT. It acts as a greater predictive significance correlated to the different expressions of proteins [39]. CK19, keratin cells, are found normally in the basal cells of the epithelium of oral squamous [40]. They are found to be irregular and augmented in OSCC. The overexpression of CK19 is one of the best biomarkers in the early diagnosis of OSCC [41,42]. Some studies show that overexpression of CK13 and CK17 in precancerous lesions of the oral region with the epithelium (atypical) are found in the OSCC progression marker [43,44]. The other cytoskeleton proteins, such as acetylated tubulin, S100A4, and β III-tubulin, are involved in EMT, which acts as an OSCC potential therapeutic target [45,46,47].

## 6. Transcription Factors Induced by EMT

### 6.1. Family—Twist

Different reports show the twisted role of carcinogenesis in the oral region and OSCC progression. In the family of Twist, there are two members, such as Twist 1 and Twist 2, belonging to the loop-helix and basic helix family. In OSCC, the Twist nuclear localization is highly correlated to the attenuated expression of E-cadherin and the augmented level of N-cadherin [48]. The overexpression of Twist is highly involved in the metastasis of cancer and not encouraging in OSCC prognosis [49]. The Twist expression is highly associated with clinical lymph node metastasis, which is involved in carcinogenesis and OSCC progression [50].

### 6.2. Family—Snail

The Slug and Snail are the two important regulators involved in EMT via different signaling pathways. Activation of these families is highly related to the metastasis of cancer. Snail is one of the E-cadherin transcriptional repressors. Slug and Snail would attenuate the epithelial marker expression via occludins and claudins [51]. In the nucleus, there is an elevated expression and accretion of Slug, and Snail attenuates the epithelial markers. The increased expression of TGF-β increases the expression of Snail and Slug. It mediates the anti-tumor drug resistance in OSCC. The Snail and Slug act together as therapeutic targets for oral cancer. It also has resistance to different therapies involved in OSCC [52].

### 6.3. ΔNp63

p63 has various isoforms, such as a lack of p63 (ΔNp63) and transactivation domain p63 (TAp63). The suppressor of the tumor is TAp63, and the oncoprotein is ΔNp63 [53]. Some studies show that ΔNp63 is highly involved in the progression of tumors. It is highly related to the tumor grade and OSCC advanced stage [54]. A change in the expression of ΔNp63 is highly associated with the dysplasia of the oral epithelium and participates in a vital role in epithelial cells in the transformation of malignancy [55]. The downregulation of the ΔNp63 in oral cancer cells promotes OSCC progression by inducing a mesenchymal phenotype [56].

### 6.4. Family—ZEB

In epithelial cells, there is an augmented level of ZEB2 or ZEB1, which mediates EMT via downregulation of some of the epithelial markers and expression of E-cadherin. Mesenchymal gene expression, such as N-cadherin, matrix metalloproteinases, and vimentin, is augmented along with the increased level of proteins of ZEB upon EMT [57]. In OSCC cells, there is an upregulation of ZEB-1, and it is often found in the nucleus. It acts as one of the predictive biomarkers related to other parameters and poor projection of OSCC [58]. The increased expression of ZEB-1 and decreased E-cadherin contribute to poor survival from oral cancer [58].

### 6.5. Other Transcriptional Factors

In OSCC, there is an augmentation of E12/E47 that induces EMT via downregulating the expression of E-cadherin [59]. Transcriptional factors such as NANOG, OCT4, and SOX2 are biomarkers for OSCC. The attenuated expression of SOX2 is highly associated with carcinogenesis in the oral cavity and OSCC progression [60] (Table 2).

## 7. Transcription Factors Inhibited by EMT

The molecular mechanisms behind EMT inhibition are not completely elucidated. The transcription factor GRHL2 triggers Claudin-4 and E-cadherin, which has a GRHL2 feedback loop. The augmentation of the GRHL2 or OVOL2 is not sufficient to mediate MET. Their roles in oral cancer need to be elucidated [61,62]. OVOL1/2 transcriptional factors can impress ZEB-1 transcriptional factors associated with EMTvia a feedback loop and regulate the splicing of mRNA via epithelial splicing regulatory protein 1 (ESRP1) induction in oral cancer. The increased expression of OVOL1/2 or GRHL2 is not essential for the complete MET process [3].

## 8. Receptors Induced by EMT during Oral Cancer

### 8.1. Tyrosine Kinase Receptor and EMT during Oral Cancer

#### 8.1.1. Fibroblast Growth Factor Receptor (FGFR) and EMT during Oral Cancer

TheFGFR is one of the subfamilies of the receptor tyrosine kinases; it has five members. This receptor is activated after the binding of FGFs in the domain of the extracellular region. It activates the downstream signaling in the intracellular region [63]. Dysregulation of FGFR signaling is often found in various cancer types and oral cancer, leading to augmented expression, abnormal isoform splicing, subcellular location, and mutation [64]. In OSCC cells, autocrine signaling such as FGF–FGFR (IIIc) is highly involved in maintaining the expression of ZEB1/2 and its phenotype of EMT. FGFRs induced by EMT are highly implicated in forming oral cancer [65].

#### 8.1.2. Epidermal Growth Factor Receptor (EGFR) and EMT during Oral Cancer

The EGFR is one of the subfamilies of the receptor tyrosine kinases; it has four members. There is a transmission of signals which is consequential from the necessary members of the EGFR family, which further causes different changes in the physiological system [66,67]. In many studies, there is an activation of EMT by EGFR through the signaling of Integrin/EGFR–ERK/MAPK. It further attenuates the ZO-1, E-cadherin, β-catenin, and cell morphology transformation [68]. The increased expression of EGFRs is highly correlated with precancerous lesions in the oral region. It is involved in transforming pathological conditions and the progression of malignant oral epithelium [68,69]. In OSCC, there is an elevated expression of EGFRs, and it is highly associated with cancer metastasis and resistance to drug-mediated EMT [70].

#### 8.1.3. Ephrin (Ephs) Receptor and EMT during Oral Cancer

Ephs is one of the subfamilies of receptor tyrosine kinases. Ephs are straightforwardly linked to metastasis and tumorigenesis. In OSCC, there is an increased expression of Ephs [71]. In OSCC, the augmented level of the Ephs is highly associated with the clinical stage, tumor size, recurrence, lymph invasion, and metastasis. It is determined as one of the self-regulating poor diagnostic factors in OSCC. It may be utilized as a diagnostic marker [72].

#### 8.1.4. Other Tyrosine Kinase Receptors and EMT during Oral Cancer

Apart from the FGFRs, Ephs, and EGFRs of the tyrosine kinase family, the family of the insulin receptor, VGEF receptor, and the receptor of PDGF are often found in the tumorigenesis and generation of OSCC. Nevertheless, in OSCC, there is an unclear subfamily mechanism [3].

### 8.2. G-Protein Coupled Receptor and EMT during Oral Cancer

#### 8.2.1. C-X-C Chemokine Receptors and EMT

CXCR4 is one of the receptors in alpha chemokine. It is mainly triggered by the stromal-derived factor-1. Additionally, extracellular ubiquitin acts as an agonist for its activation [73]. In OSCC, CXCR4 acts as one of the receptors of chemokines, which is highly involved in cancer metastasis. There is an increased level of the CXCR4 receptor in the lymph nodes and tumor cells which is linked to metastasis, tumorigenesis, and drug resistance. It contributes to the poor outcome of OSCC patients [73]. There is a generation of the migration of cells and invasion, which is highly related to the contribution of EMT via the augmented expression of vimentin, Snail, and matrix metalloproteinases [74]. The OSCC metastasis is regulated by other chemokines receptors such as CCR7, CCR4, CCR5, and CXCR7. It is highly associated with the diagnosis of OSCC patients [75].

#### 8.2.2. Histamine Receptors and EMT

One of the main chemical transmitters is histamine, which shows its function via receptor binding such as H1R to H4R via G-protein coupled action. The immune action of the H4R is highly attenuated in OSCC and the dysplasia of oral epithelial cells [76]. Though the expression of the H1R is very rare in OSCC, its increased expression is linked with a poor prediction of OSCC [77]. The activation of the histamine receptor plays an important role in the carcinogenesis of OSCC (Table 3 and Figure 1).

## 9. Other Signaling Events Induced by EMT during Oral Cancer

### 9.1. PI3 Kinase/mTOR/Akt Signaling and EMT with Oral Cancer

TGF-β activates PI3/Akt signaling, which plays a vital role in EMT induction. Akt is linked to increased proliferation, invasiveness, anti-apoptosis, and growth. Akt enhancement is closely linked with the poor diagnosis of OSCC patients [78]. There is a mutation of the gene involved in the pathways, such as Akt, PI3, PTEN, and RAS [79].

### 9.2. Wnt Signaling in Oral Cancer and EMT

Secreted frizzled-related protein, Wnt inhibitory factor 1, and families of Dickkopf act as antagonists for the signaling pathway such as Wnt/β-catenin. In OSCC patients, there is a DKK3 mutation. The Wnt antagonist, WIF1, is decreased in OSCC because of its methylation. The methylation of WIF1 is closely related to the poor diagnosis of OSCC. WIF1 is an indicator of OSCC, which acts as a biomarker for epigenetics [80]. The signaling of Wnt/2-catenin activation shows an expression of LEF1 and β-catenin in OSCC patients [81]. There is no change in the downstream molecules of this pathway in OSCC patients except for c-MYC [82]. There is a β-catenin transient expression in OSCC patients due to the different mechanisms [83].

### 9.3. Matrix Signaling and EMT in Oral Cancer

Augmentation of MMP1, MMP7, MMP9, and MMP2 is determined in the OSCC, which plays an important role in the degradation of the extracellular matrix and basement membrane. It is highly related to clinical significance, such as metastasis and lymph node in the regional [84]. Collagen type 1 binds with the αvβ8 integrin and mediates the ERK/MEK pathway by mediating OSCC generation and invasions. αvβ8 expression is necessary for the proliferation of cells, migration, and adhesion in OSCC progression [85].

### 9.4. Notch Signaling Pathway and EMT in Oral Cancer

The mutation of Notch is found in the squamous cell carcinoma of the oral cavity. An impairment of Notch causes epidermis hyperplasia and elevates the incidence of tumors in theearly stages of OSCC [86]. EMT activates Notch signaling via elevated Snail and reduced expression of E-cadherin, which mediates tumor metastasis [87]. Therefore, Notch, an oncogene in OSCC, also suppresses tumors [88].

### 9.5. Hedgehog Signaling and EMT in Oral Cancer

The sonic (SHH) hedgehog homolog shows an enhancement in OSCC and acts as a therapeutic target for oral cancer. A prospective promoter, such as SHH signaling, is highly associated with OSCC [89,90]. In OSCC, activating its signaling induces metastasis and tumorogenesis via EMT [91].

**Table 3 vaccines-10-01490-t003:** **Receptors induced by EMT during oral cancer**.

S.No.	Receptors	Genes Involved	Outcome	References
1.	1.1. Tyrosine kinase receptor 1.1.1. Fibroblast growth factor receptors (FGFRs) Increased FGFRs 1.1.2. Epidermal growth factor receptor (EGFR) Increased EGFR Elevated Integrin/EGFR-ERK/MAPK 1.1.3. Ephrin receptor Increased Ephs 1.1.4. Other tyrosine kinase receptors Increased Insulin receptor, VGEF, and PDGF receptor 1.2. G-protein coupled receptor 1.2.1. C-X-C chemokine receptors Increased stromal-derived factor 1 1.2.2. Histamine receptor Increased H1R	Increased ZEB1/2 Decreased ZO-1, E-cadherin, β–catenin Increased Ephs - Augmented level of vimentin, Snail, and matrix metalloproteinases Increased H1R	EMT induction EMT induction Cell morphology transformation Increased EMT Increased tumor size, recurrence, lymph invasion, and metastasis Increased tumorigenesis EMT induction Migration of cells and invasion Elevated OSCC generation	[65] [70] [71,72] [3] [75] [77]
2.	2.1. PI3 kinase/mTOR/Akt signaling TGF-β activates PI3/Akt signaling	Mutation in Akt, PI3, PTEN, and RAS	Increased proliferation, invasiveness, anti-apoptosis, and growth	[78,79]
3.	3.1. Wnt signaling Decreased Wnt antagonist, WIF1	Increased LEF1 and β-catenin	EMT induction and increased oral cancer progression	[82,83]
4.	4.1. Matrix signaling Increased MMP1, MMP7, MMP9, and MMP2, and collagen type 1	Elevated ERK/MEK level	Degradation of the extracellular matrix and basement membrane	[85]
5.	5.1. Notch signaling	Increased Snail and reduced expression of E-cadherin	Increased tumor metastasis	[87,88]
6.	6.1. Hedgehog signaling SHH enhancement	-	Increased tumorigenesis via EMT induction	[89,90]
7.	7.1. Phosphatase and tensin homolog deleted on chromosome 10 (PTEN)	Decreased PTEN	Cancer invasion via EMT induction	[6]

### 9.6. Phosphatase and Tensin Homolog Deleted on Chromosome 10 (PTEN) and EMT in Oral Cancer

PTEN impairment or its loss mediates the progression of malignancy. In OSCC, there is a decreased level of PTEN compared to normal cells. The upregulation of PTEN decreases cancer invasion through EMT inhibition [6] (Figure 1 and Table 3).

### 9.7. Neuropilin-1 (NRP1) and EMT in Oral Cancer

Neuropilin-1 (NRP1) is often in oral cancer cells. The increased expression of neuropilin-1 (NRP1) contributes to the migration of cells and invasion of oral cancer cells. NF-kB plays an important role in EMT activation and its maintenance. Further, it is involved in tumor metastasis. NRP-1 mediates EMT via cancer cell migration and invasion. This is highly involved in the progression of OSCC invasion and metastasis via NF-kB activation [92].

### 9.8. Pituitary Tumor-Transforming Gene 1 (PTTG1) and EMT in Oral Cancer

PTTG1 plays an important role in the transcription of oral malignancies. There is an increased expression of PTTG1 in the OSCC tissues. The increased expression of PTTG1 contributes to the increased level of the MMPs, especially MMP-2 and further causes elevated levels of EMT. This modification in EMT downregulates the level of E-cadherin and augmentation of the Snail and vimentin. This further contributes to oral cancer progression [93].

### 9.9. Transforming Growth Factor-β 1-Activated Kinase 1 Binding Protein 2 (TAB2) and EMT in Oral Cancer

Transforming growth factor-β 1-activated kinase 1 binding protein 2 (TAB2) is highly involved in various processes in cancer progression via NF-kB signaling pathways. The expression of TAB2 is high in oral cancer. It is highly associated with the diagnosis of oral cancer. The impairment of TAB2 leads to reduced progression and elevated apoptosis. This regulates OSCC generation via PI3K–AKT signaling pathways and EMT enhancement [94].

### 9.10. Engulfment and Cell Motility (ELMO) Proteinsand EMT in Oral Cancer

The impairment of the TGFβ receptor II (Tgfbr2) causes cancer cell metastasis and invasion via ELMO1 repression. ELMO1 is a guanine exchange factor via RAC. ELMO1 acts as a proper target for the TGFβ and restores the activities of the Tgfbr2. The impairment of ELMO1 decreases the metastasis of the cancer cells [95].

## 10. MicroRNAs and EMT in Oral Cancer

MicroRNAs act as the main EMT regulation in OSCC. In OSCC, the primary microRNAs determined are miR-200. The reduction in the family of miR-200 mediates the migration of malignant cells. Thus, it elevates the EMT level via the augmented level of the gene expression of mesenchymal cells. Along with that, it suppresses the transcriptional factors associated with EMT. The inhibition of miR-205, miR-488, mir-485-5p, and miR-204 enhances EMT and other biological activity via signaling pathways in OSCC cells. In contrast, miR-19a, miR-221, miR-155-5p, and miR-424 are often found in OSCC cells and are associated with poor diagnosis [6].

## 11. Oral Cancer—EMT Regulation via Microenvironment

### 11.1. Oral Cancer and Fibroblasts

Fibroblasts associated with oral cancer are related to less differentiation and the migration of cells in OSCC patients [5]. Fibroblasts generate tumor metastasis by changing the biological activities in oral cells. Fibroblast-associated oral cancer is determined by the changes in markers of mesenchymal and epithelial cells at both the protein and RNA levels. It further mediates the metastasis of OSCC via EMT generation [6].

### 11.2. Integrins

Integrins induce the communication between EMT and neoplastic cells through the association with the transmembrane, which is necessary for OSCC metastasis [6].

## 12. Conclusions

In this review article, we conclude that in OSCC patients, there is an alteration in immunological action via the tumor microenvironment and EMT. There is an alteration in the inflammatory markers such as TGF-β, TNF-α, IL-1β, IL-6, IL-8, MCP-1/CCL2, and macrophages. EMT mediates tumor hypoxia and cytoskeleton changes. EMT induces transcription factors such as the Twist family, Snail family, ZEB family, ΔNp63, and other transcriptional factors. It also alters the tyrosine kinase receptor family and GPCR family and other signaling events by EMT in oral cancer. From this, we conclude that EMT and the tumor microenvironment with immunological changes act as diagnostic markers for OSCC.

## Figures and Tables

**Figure 1 vaccines-10-01490-f001:**
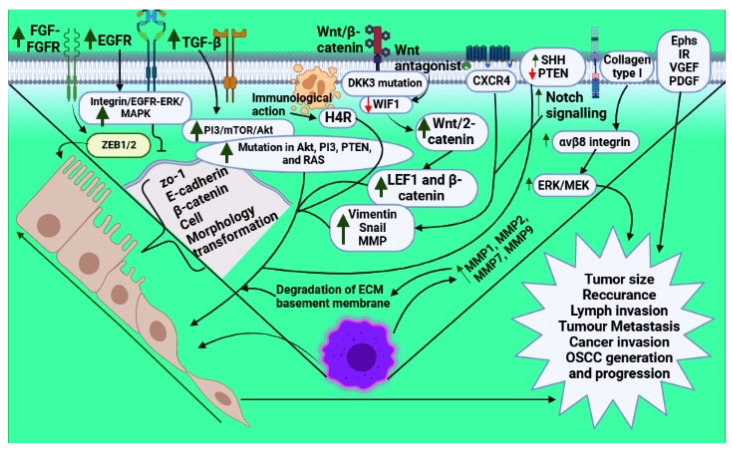
**Figure 1 shows the signaling events involved in EMT during oral cancer via different receptors**.

**Table 1 vaccines-10-01490-t001:** **Role of Inflammatory proteins which influence EMT in oral cancer**.

S.No.	Inflammatory Proteins	Factors/Genes Involved	Outcome	References
1.	1.1. Transforming growth factor β (TGF-β) in oral cancer 1.1.1. Augmented Smad and non-Smad signaling 1.1.2. Elevated THBS-2 via TGF-β 1.1.3. BMP-2 to BMP-7	Decreases E-cadherin and increases vimentin Increases MMPs Decreases Snail; N-cadherin and CK9	EMT induction EMT induction EMT induction; increases tumor differentiation, lymph node metastasis	[13,14,15,16]
2.	2.1 Tumor necrosis factor-alpha (TNF-α) and oral cancer 2.1.1. Increased TNF-α	Increases MAPK level	EMT augmentation Increased mesenchymal marker Decreased epithelial marker	[17,18,19,20]
3.	3.1 Interleukin-1β (IL-1β) role in oral cancer 3.1.1. Increased IL-1β	Increased NF-kB; AP-1; IL-8; GROα, IL-6	EMT induction	[22]
4.	4.1 Interleukin-6 (IL-6) in oral cancer 4.1.1. Increased IL-6	Activation of JAK/STAT3	EMT induction	[24,25]
5.	5.1 Interleukin-8 (IL-8) and oral cancer 5.1.1. Increased IL-8	Increased p38 and MAPK kinase	EMT activation E-cadherin epigenetic silencing	[26]
6.	6.1 Monocyte chemoattractant protein-1(MCP-1/CCL2) and oral cancer 6.1.1. CCL2 activation		EMT activation	[29]
7.	7.1 Macrophage and oral cancer 7.1.1. Increased macrophages	Decreased ZO-1 and E-cadherin (epithelial markers) Increased vimentin and N-cadherin (mesenchymal markers)	Increased EMT Increased tumor metastasis	[30,31]

**Table 2 vaccines-10-01490-t002:** Transcription factors induced by EMT.

S.No.	Transcription Factor	Genes Involved	Outcome	References
1.	Family—Twist Increased Twist 1 and Twist 2	Decreased E-cadherin and increased N-cadherin	Increased EMT level Increased metastasis of cancer	[48,49,50]
2.	Family—Snail Increased Snail	Decreased E-cadherin, occludins, and claudins	Decreased epithelial markers EMT activation	[51]
3.	ΔNp63 Augmented ΔNp63	-	Increased dysplasia Increased transformation of malignancy	[54]
4.	Family—ZEB Increased ZEB2 or ZEB1	Decreased E-cadherin and increased N-cadherin; MMPs; vimentin; ZEB	EMT induction	[57,58]
5.	Other transcriptional factors Increased E12/E47 Increased SOX2	Decreased E-cadherin and SOX2	Increased OSCC progression	[60]

## Data Availability

Data are available from the authors on request (A.V.G.) and (K.R.).

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
