# Peer review of "Molecular Crosstalk between the Immunological Mechanism of the Tumor Microenvironment and Epithelial–Mesenchymal Transition in Oral Cancer"

_vaccines, 2022, doi:10.3390/vaccines10091490_

Round 1
Reviewer 1 Report
Renu and colleagues presented a review article on the molecular crosstalk between Tumor Microenvironment (TME) and Epithelial-Mesenchymal Transition (EMT) in oral cancer. Specifically, the authors focused on inflammatory proteins, transcription factors, receptors, different signaling pathways, and miRNAs investigating their role in oral cancer. The researchers highlighted that the alteration of immunological action via TME and EMT could represent a diagnostic marker for oral cancer patients. The manuscript is well supported by consistent references and encourages further studies to better understand the mechanism behind TME and EMT during oral cancer. However, I have some major comments as follows:
1) The scientific background and the purpose of the review should be better explained in the Introduction section.
2) Section 2 “Epithelial-mesenchymal transition in oral cancer” should be better described. Specifically, what is reported in the literature regarding the role of EMT in oral cancer?
3) Section 3 “Inflammatory proteins’ role in influencing EMT in oral cancer” should be implemented with a summery table describing the inflammatory markers cited in the main text. Moreover, a similar table should be also added in Sections 6, 8, and 9 to provide a comprehensive view of the transcription factors, receptors, and other signaling events involved in oral cancer.
4) The content of some subparagraphs should be enriched with more details and new studies (i.e. from 3.2 to 3.7; from 6.1 to 6.5; 9.5 and 9.6; 11.1 and 11.2). Alternatively, the aforementioned subparagraphs may be unified into a single paragraph. Similarly, Section 7 “Transcription factors inhibited by EMT” should be extended adding more information.
5) A visual figure or a graphical abstract may be added to better explain the message of the manuscript in a clear and attractive way.
Author Response
Reviewer 1:
Renu and colleagues presented a review article on the molecular crosstalk between Tumor Microenvironment (TME) and Epithelial-Mesenchymal Transition (EMT) in oral cancer. Specifically, the authors focused on inflammatory proteins, transcription factors, receptors, different signaling pathways, and miRNAs investigating their role in oral cancer. The researchers highlighted that the alteration of immunological action via TME and EMT could represent a diagnostic marker for oral cancer patients. The manuscript is well supported by consistent references and encourages further studies to better understand the mechanism behind TME and EMT during oral cancer. However, I have some major comments as follows:
Response: The authors thank to the Reviewer for the favourable evaluation of our work and for the specific points raised for improvement of the manuscript.
1) The scientific background and the purpose of the review should be better explained in the Introduction section.
Response 1: According to the suggestion of reviewer 1 we have modified in the introduction section.
It is known to process that EMT is a dynamic reversible process that mediates the epithelial cells to undergo mesenchymal cells via different biochemical changes, increased capacity of migration, apoptotic resistance, invasiveness, and increased ECM production [3]. Although there is a study shows that EMT is not necessary for the metastasis of the tumor but it is highly involved in cancer metastasis and invasion. Currently, different studies are showing that the tumor cells involved in the circulation and cell migration cause tumor metastasis which further leads to EMT [3]. Therefore, we could consider EMT acts as a hotspot in the research of cancer and it is highly involved in metastasis which gives the main principle for oncothereapy. The goal of our study is to give the overall view of EMT in Oral cancer with different signaling mechanisms, transcriptional factors, etc., This review focuses on the factors mediating TME, the subpopulation of cells with immunological aspects, epithelial-mesenchymal cell transition, and its function in oral cancer development.
2) Section 2 “Epithelial-mesenchymal transition in oral cancer” should be better described. Specifically, what is reported in the literature regarding the role of EMT in oral cancer?
Response 2: We have satisfied the comment.
At the level of molecular, there are 3 mechanisms involved importantly for the EMT in oral cancer such as 1. Decreased level of the E cadherin (the main thing in EMT in oral cancer is the adhesion of cell-cell and elevated motility of the cells via E-cadherin downregulation and N-cadherin upregulation); 2. microRNA expression modification (deregulation of the miRNA has been highly connected to the metastasis and resistance to the tumor by changing EMT in oral cancer); 3. Actin reorganization and invadopodia formation (this intracellular filament of actin is reorganized via β catenin and E-cadherin adhesion and MMPs are involved in the formation of invadopodia in oral cancer) [5].
3) Section 3 “Inflammatory proteins’ role in influencing EMT in oral cancer” should be implemented with a summery table describing the inflammatory markers cited in the main text. Moreover, a similar table should be also added in Sections 6, 8, and 9 to provide a comprehensive view of the transcription factors, receptors, and other signaling events involved in oral cancer.
Response 3: We have satisfied the comment.
Table 1: Role of Inflammatory proteins' which influence EMT in oral cancer
|
S.No |
Inflammatory proteins |
Factors/genes involved |
Outcome |
Reference |
|
1. |
1.1.Transforming Growth Factor β (TGF-β) in oral cancer 1.1.1. Augmented smad and non-smad signaling 1.1.2. Elevated THBS-2 via TGF-β 1.1.3. BMP-2 to BMP-7 |
Decreases E-cadherin and increases vimentin Increases MMPs Decreases Snail; N-cadherin and CK9 |
EMT induction
EMT induction EMT induction; increases tumor differentiation, lymph node metastasis |
[13-16] |
|
2. |
2.1.Tumor Necrosis Factor - Alpha (TNF-α) and oral cancer 2.1.1. Increased TNF- α |
Increases MAPK level |
EMT augmentation Increased mesenchymal marker Decreased epithelial marker |
[17-20] |
|
3. |
3.1.Interleukin-1β (IL-1β) role in oral cancer 3.1.1. Increased IL-1β |
Increased NF-kB; AP-1; IL-8; GROα, IL-6 |
EMT induction |
[22] |
|
4. |
4.1.Role of Interleukin-6 (IL-6) in oral cancer 4.1.1. Increased IL-6 |
Activation of JAK/STAT3 |
EMT induction |
[24-25] |
|
5. |
5.1.Interleukin-8 (IL-8) and oral cancer 5.1.1. Increased IL-8 |
Increased p38 and MAPK kinase |
EMT activation E-cadherin epigenetic silencing |
[26] |
|
6. |
6.1.Monocyte chemoattractant protein-1(MCP-1/CCL2) and oral cancer 6.1.1. CCL2 activation |
|
EMT activation
|
[29] |
|
7. |
7.1.Macrophage and oral cancer 71.1. Increased macrophages |
Decreased ZO-1 and E-cadherin (epithelial markers) Increased vimentin and N-cadherin (mesenchymal markers) |
Increased EMT Increased tumor metastasis |
[30-31] |
Table 2: Transcription factors induced by EMT
|
S.No. |
Transcription factor |
Genes involved |
Outcome |
Reference |
|
1. |
Family- Twist Increased Twist 1 and Twist 2 |
Decreased E-cadherin and increased N-cadherin |
Increased EMT level Increased metastasis of cancer |
[48-50] |
|
2. |
Family – Snail Increased Snail |
Decreased E-cadherin, occludins, and claudins |
Decreased epithelial markers EMT activation |
[51] |
|
3. |
Δ. Np63 Augmented Δ. Np63 |
- |
Increased dysplasia Increased transformation of malignancy |
[54] |
|
4. |
Family – ZEB Increased ZEB2 or ZEB1 |
Decreased E-cadherin and increased N-cadherin; MMPs; Vimentin; ZEB |
EMT induction |
[57,58] |
|
5 |
Other transcriptional factors Increased E12/E47 Increased Sox2 |
Decreased E-cadherin and Sox2 |
Increased OSCC progression |
[60] |
Table 3: Receptors induced by EMT during oral cancer
|
S.No. |
Receptors |
Genes involved |
Outcome |
Reference |
|
1. |
1.1.Tyrosine kinase receptor 1.1.1. Fibroblast growth factor receptors (FGFRs) Increased FGFRs 1.1.2. Epidermal growth factor receptor (EGFR) Increased EGFR Elevated Integrin/EGFR-ERK/MAPK
1.1.3. Ephrin receptor Increased Ephs
1.1.4. Other tyrosine kinase receptors Increased Insulin receptor, VGEF, and PDGF receptor 1.2.G-protein coupled receptor 1.2.1. C-X-C chemokine receptors Increased stromal-derived factor 1 1.2.2. Histamine receptor Increased H1R
|
Increased ZEB1/2
Decreased ZO-1, E-cadherin, β –catenin
Increased Ephs
-
Augmented level of vimentin, Snail, and matrix metalloproteinases
Increased H1R |
EMT induction
EMT induction Cell morphology transformation Increased EMT
Increased tumor size, recurrence, lymph invasion, and metastasis
Increased tumorigenesis
EMT induction Migration of cells and invasion
Elevated OSCC generation |
[65]
[70]
[71,72]
[3]
[75]
[77]
|
|
2. |
2.1. PI3 kinase/mTOR/Akt signaling TGF- β activates PI3/Akt signaling |
Mutation in Akt, PI3, PTEN, and RAS |
increased proliferation, invasiveness, anti-apoptosis, and growth |
[79] |
|
3. |
3.1. Wnt signaling Decreased Wnt antagonist - WIF1 |
Increased LEF1 and β-catenin |
EMT induction and increased oral cancer progression |
[82,83] |
|
4. |
4.1. Matrix signaling Increased MMP1, MMP7, MMP9, and MMP2 and Collagen type 1 |
Elevated ERK/MEK level |
Degradation of the extracellular matrix and basement membrane |
[85] |
|
5. |
5.1. Notch signaling |
Increased Snail and reduced expression of E-cadherin |
Increased tumor metastasis |
[87,88] |
|
6. |
6.1. Hedgehog signaling SHH enhancement |
- |
Increased tumorigenesis via EMT induction |
[89,90] |
|
7. |
7.1. Phosphatase and tensin homolog deleted on chromosome 10 (PTEN) |
Decreased PTEN |
Cancer invasion via EMT induction |
[6] |
4) The content of some subparagraphs should be enriched with more details and new studies (i.e. from 3.2 to 3.7; from 6.1 to 6.5; 9.5 and 9.6; 11.1 and 11.2). Alternatively, the aforementioned subparagraphs may be unified into a single paragraph. Similarly, Section 7 “Transcription factors inhibited by EMT” should be extended adding more information.
Response 4: We have incorporated the changes in the revised manuscript.
3.2. The connection between Tumor Necrosis Factor - Alpha (TNF-α), Interleukin-1β (IL-1β) , Interleukin-6 (IL-6) , Interleukin-8 (IL-8) , Monocyte chemoattractant protein-1(MCP-1/CCL2) , Macrophage and oral cancer
Like TGF-β, TNF-α plays a twin role in malignant tumors. TNF-α can destroy tumor cells (OSCC) due to its response to immune and inflammation. TNF-α generates inva-siveness of cancer and metastasis of cancer through triggering signaling pathways like MAPK, which induces EMT [17]. In OSCC, TNF- α elevates the mesenchymal marker expression and attenuates the epithelial marker expressions [18]. Some studies show that TNF- α impedes the activity of OSCC migration [19]. TNF-α and TNFR1 play an important role in the progression of OSCC. The neutralization of TNF-α reduces the cytokines in serum which inhibits the invasive lesion progressions and further decreases the neu-trophils associated with the tumor in vivo. This shows the role of TNF-α in the trans-formation of oral malignant via regulating TNFR1. This acts as a diagnostic feature for OSCC [20].
In the immune cells, IL-1β is triggered by the transcription factors such as NF-κB and AP-1. IL-1β is highly expressed in the stromal cells of tumors and effectors cells which infiltrate the tumor immune system. It is highly responsible for tumor microenvironment shaping. In the OSCC and oral keratinocytes, IL-1β strengthens EMT via proinflammatory cytokine production like IL-8, GROα, and IL-6 [21]. The expression of the oral epithelial IL-1β was found to be decreased in increased oral mucosa malignant transformation [22].
IL-6 is one of the multifunctional cytokines which is found highly in different tissues and acts in a paracrine or autocrine manner. There is a relation between the survival of cancer tumors, metastasis of tumor, relapse, OSCC therapeutic resistance, and IL-6 [23]. Though there are different mechanisms of IL-6, the exact mechanism behind cancer pro-gression remains unclear. EMT activation by IL-6 via JAK/STAT3 activation plays an important role in the generation of OSCC [24]. There is an increased IL-6 level in the saliva of OSCC patients compared to normal patients [25].
IL-8 triggers signaling pathways such as JAK/STAT. PI3K and Ras, MAPK, and Raf via binding with the GPCR like CXCR1/2. Erlotinib is a target for EGFR tyrosine kinase, which triggers off the IL-8 secretion associated with the EMT via the p38 and MAPK ki-nase pathway. In some cases, such as nasopharyngeal carcinoma, EMT activates via dif-ferent pathways such as E-cadherin epigenetic silencing [26]. The expression of the oral epithelial IL-8 was found to be decreased in increased oral mucosa malignant transfor-mation [22].
CCL2 is a cytokine with different types of immune cells to cause acute inflammation. CCL2 binds with the receptors CCR2 and CCR4 to produce a diverse response in the bi-ological system [27]. Along with that, it also binds with the other receptors such as ACKR2 and ACKR1. So, this CCL2 and its receptor have an important role in inflammation. More evidence shows that CCL2 is associated with cancer metastasis in OSCC. It triggers many signaling pathways to mediate EMT [28]. The increased expression of the MCP-1 mediates the progression of OSCC via increasing the signaling pathways involved in pro-survival. The increased expression of the MCP-1 leads to a decreased survival rate in oral cancer patients [29].
The macrophages are associated with the tumor, generating many mediators that regulate the tumor’s biological activity. Tumor-associated macrophages trigger the M1 (classical) or M2 (alternative) polarization phenotype, which has an activity opposite in nature, such as pro-tumor activity and anti-tumor activity [30]. The tumor-associated macrophage accumulations are closely related to the poor outcome in clinical patients. There is an attenuation of the ZO-1 and E-cadherin (epithelial markers) and augmentation of the vimentin and N-cadherin (mesenchymal markers) in tumor-associated ma-crophages in OSCC cells. The transcription factors induced by EMT are Slug and Twist, which are elevated in tissues of OSCC. Macrophages are one of the important inflam-matory cells which play an important role in the progression of oral cancer. This is in-volved in the oral cancer development, oral precancerous lesions, etc., This accumulation of tumor-associated macrophages could cause tumor metastasis to the EMT activation [31] (Table 1).
5) A visual figure or a graphical abstract may be added to better explain the message of the manuscript in a clear and attractive way.
Response 5: We have added the graphical abstract.
Figure 1: Figure 1 shows the signaling events involved in the EMT during oral cancer via different receptors.
We thank the reviewers for their valuable suggestions to highlight the significance of this review and improve the review article to meet the journal’s standards. We hope that this revised manuscript is now better suited for publication in your esteemed journal.

Reviewer 2 Report
The above manuscript is a literature review article. Therefore, there are no Methods or Discussion for revision. As for its contents, this review focuses on the factors influencing the immunological mechanism of the tumor microenvironment and epithelial-mesenchymal transition in oral cancer. I believe that most of these factors are listed and properly reviewed in this manuscript. I would also propose to the authors to include and review in their manuscript three more factors; -Neuropilin-1 -Pituitary tumor-transforming gene 1 and -TAB2 All these factors are recently reported to promote epithelial-mesenchymal transition in oral cancer.Although it is not original, it's a nice and well-written review concerning the important field of epithelial-mesenchymal transition in oral cancer. References are appropriate and up to dated. If this topic falls into the scientific spectrum and interests of Vaccines Journal, then I recommend it for publication.
Author Response
Reviewer 2:
Although it is not original, it's a nice and well-written review concerning the important field of epithelial-mesenchymal transition in oral cancer. References are appropriate and up to dated. If this topic falls into the scientific spectrum and interests of Vaccines Journal, then I recommend it for publication.
Comments and Suggestions for Authors
The above manuscript is a literature review article. Therefore, there are no Methods or Discussion for revision. As for its contents, this review focuses on the factors influencing the immunological mechanism of the tumor microenvironment and epithelial-mesenchymal transition in oral cancer. I believe that most of these factors are listed and properly reviewed in this manuscript.
Response: The authors thank to the Reviewer for the favourable evaluation of our work and for the specific points raised for improvement of the manuscript.
1: I would also propose to the authors to include and review in their manuscript three more factors; -Neuropilin-1 -Pituitary tumor-transforming gene 1 and -TAB2 All these factors are recently reported to promote epithelial-mesenchymal transition in oral cancer.
Response 1: We have satisfied the comment.
9.7. Neuropilin-1 (NRP1) and EMT in oral cancer
Neuropilin-1 (NRP1) is highly found in oral cancer cells. The increased expression of neuropilin-1 (NRP1) contributes to the migration of cells and invasion of oral cancer cells. NF-kB plays an important role in the EMT activation and maintenance of it. Further, it is involved in tumor metastasis. The NRP-1 mediates EMT vial cancer cell migration and invasion. This is highly involved in the progression of the OSCC invasion and metastasis via NF-kB activation [92].
9.8. Pituitary tumor-transforming gene 1 (PTTG1) and EMT in oral cancer
PTTG1 plays an important role in the transcription of oral malignancies. There is an increased expression of PTTG1 in the OSCC tissues. The increased expression of PTTG1 contributes to the increased level of the MMPs especially MMP-2 and further causes elevated levels of the EMT. This modification in the EMT downregulates the level of the E-cadherin and augmentation of the snail and vimentin. This further contributes to oral cancer progression [93].
9.9. Transforming growth factor β 1-activated kinase 1 binding protein 2 (TAB2) and EMT in oral cancer
Transforming growth factor β 1-activated kinase 1 binding protein 2 (TAB2) is highly involved in various processes in cancer progression via NF-kB signaling pathways. The expression of TAB2 is high in oral cancer. It is highly associated with the diagnosis of oral cancer. The impairment of the TAB2 leads to reduced progression and elevated apoptosis. This regulates the OSCC generation via PI3K-AKT signaling pathways and EMT enhancement [94].
We thank the reviewers for their valuable suggestions to highlight the significance of this review and improve the review article to meet the journal’s standards. We hope that this revised manuscript is now better suited for publication in your esteemed journal.

Reviewer 3 Report
It is an interesting review discusses the immunological mechanism of the tumor microenvironment and epithelial-mesenchymal transition in oral cancer. The review includes the following points: a)epithelial-mesenchymal transition in oral cancer. b) Inflammatory proteins’ role in influencing EMT in oral cancer including TGF-B, TNF-a,IL-1B, IL-6, IL-8, and MCP-1. c) EMT induced Tumor hypoxia and oral cancer. d) Alteration of cytoskeleton involved in EMT acts as a diagnostic marker for oral cancer therapy.
e) Transcription factors induced by EMT including Snail, Twist,ZEB, ..etc. In addition, transcription factors suppressed by EMT.
f) Receptors induced by EMT during oral cancer including tyrosine kinase receptors (fibroblast growth factor receptor (FGFRs), and epidermal growth factor receptor (EGFR), and Ephrin (Ephs) receptor), G-protein coupled receptor ( chemokine receptor, and histamine receptor). In additions, other signaling pathways including Wnt signaling, PI3 kinase/mTOR/Akt, and Notch signaling.
Some points to improve the quality of the review.
1- I would suggest include a table to summarize all the signaling pathways in oral cancers.
2- I also recommend including one figure that show the start and end of the signaling pathway,
3- Some pathways are missing such as ELMO1 and Smad
4-In the introduction, please include the prevalence of oral cancer, rate of death associated with oral caner, and risk factors.
Author Response
Reviewer 3:
It is an interesting review discusses the immunological mechanism of the tumor microenvironment and epithelial-mesenchymal transition in oral cancer. The review includes the following points: a) epithelial-mesenchymal transition in oral cancer. b) Inflammatory proteins’ role in influencing EMT in oral cancer including TGF-B, TNF-a, IL-1B, IL-6, IL-8, and MCP-1. c) EMT induced Tumor hypoxia and oral cancer. d) Alteration of cytoskeleton involved in EMT acts as a diagnostic marker for oral cancer therapy.
- e) Transcription factors induced by EMT including Snail, Twist, ZEB, ..etc. In addition, transcription factors suppressed by EMT.
- f) Receptors induced by EMT during oral cancer including tyrosine kinase receptors (fibroblast growth factor receptor (FGFRs), and epidermal growth factor receptor (EGFR), and Ephrin (Ephs) receptor), G-protein coupled receptor ( chemokine receptor, and histamine receptor). In additions, other signaling pathways including Wnt signaling, PI3 kinase/mTOR/Akt, and Notch signaling.
Response: The authors thank to the Reviewer for the favourable evaluation of our work and for the specific points raised for improvement of the manuscript.
Some points to improve the quality of the review.
1- I would suggest include a table to summarize all the signaling pathways in oral cancers.
Response 1: We have added the table in the main manuscript.
Table 1: Role of Inflammatory proteins' which influence EMT in oral cancer
|
S.No |
Inflammatory proteins |
Factors/genes involved |
Outcome |
Reference |
|
1. |
1.1.Transforming Growth Factor β (TGF-β) in oral cancer 1.1.1. Augmented smad and non-smad signaling 1.1.2. Elevated THBS-2 via TGF-β 1.1.3. BMP-2 to BMP-7 |
Decreases E-cadherin and increases vimentin Increases MMPs Decreases Snail; N-cadherin and CK9 |
EMT induction
EMT induction EMT induction; increases tumor differentiation, lymph node metastasis |
[13-16] |
|
2. |
2.1.Tumor Necrosis Factor - Alpha (TNF-α) and oral cancer 2.1.1. Increased TNF- α |
Increases MAPK level |
EMT augmentation Increased mesenchymal marker Decreased epithelial marker |
[17-20] |
|
3. |
3.1.Interleukin-1β (IL-1β) role in oral cancer 3.1.1. Increased IL-1β |
Increased NF-kB; AP-1; IL-8; GROα, IL-6 |
EMT induction |
[22] |
|
4. |
4.1.Role of Interleukin-6 (IL-6) in oral cancer 4.1.1. Increased IL-6 |
Activation of JAK/STAT3 |
EMT induction |
[24-25] |
|
5. |
5.1.Interleukin-8 (IL-8) and oral cancer 5.1.1. Increased IL-8 |
Increased p38 and MAPK kinase |
EMT activation E-cadherin epigenetic silencing |
[26] |
|
6. |
6.1.Monocyte chemoattractant protein-1(MCP-1/CCL2) and oral cancer 6.1.1. CCL2 activation |
|
EMT activation
|
[29] |
|
7. |
7.1.Macrophage and oral cancer 71.1. Increased macrophages |
Decreased ZO-1 and E-cadherin (epithelial markers) Increased vimentin and N-cadherin (mesenchymal markers) |
Increased EMT Increased tumor metastasis |
[30-31] |
Table 2: Transcription factors induced by EMT
|
S.No. |
Transcription factor |
Genes involved |
Outcome |
Reference |
|
1. |
Family- Twist Increased Twist 1 and Twist 2 |
Decreased E-cadherin and increased N-cadherin |
Increased EMT level Increased metastasis of cancer |
[48-50] |
|
2. |
Family – Snail Increased Snail |
Decreased E-cadherin, occludins, and claudins |
Decreased epithelial markers EMT activation |
[51] |
|
3. |
Δ. Np63 Augmented Δ. Np63 |
- |
Increased dysplasia Increased transformation of malignancy |
[54] |
|
4. |
Family – ZEB Increased ZEB2 or ZEB1 |
Decreased E-cadherin and increased N-cadherin; MMPs; Vimentin; ZEB |
EMT induction |
[57,58] |
|
5 |
Other transcriptional factors Increased E12/E47 Increased Sox2 |
Decreased E-cadherin and Sox2 |
Increased OSCC progression |
[60] |
Table 3: Receptors induced by EMT during oral cancer
|
S.No. |
Receptors |
Genes involved |
Outcome |
Reference |
|
1. |
1.1.Tyrosine kinase receptor 1.1.1. Fibroblast growth factor receptors (FGFRs) Increased FGFRs 1.1.2. Epidermal growth factor receptor (EGFR) Increased EGFR Elevated Integrin/EGFR-ERK/MAPK
1.1.3. Ephrin receptor Increased Ephs
1.1.4. Other tyrosine kinase receptors Increased Insulin receptor, VGEF, and PDGF receptor 1.2.G-protein coupled receptor 1.2.1. C-X-C chemokine receptors Increased stromal-derived factor 1 1.2.2. Histamine receptor Increased H1R
|
Increased ZEB1/2
Decreased ZO-1, E-cadherin, β –catenin
Increased Ephs
-
Augmented level of vimentin, Snail, and matrix metalloproteinases
Increased H1R |
EMT induction
EMT induction Cell morphology transformation Increased EMT
Increased tumor size, recurrence, lymph invasion, and metastasis
Increased tumorigenesis
EMT induction Migration of cells and invasion
Elevated OSCC generation |
[65]
[70]
[71,72]
[3]
[75]
[77]
|
|
2. |
2.1. PI3 kinase/mTOR/Akt signaling TGF- β activates PI3/Akt signaling |
Mutation in Akt, PI3, PTEN, and RAS |
increased proliferation, invasiveness, anti-apoptosis, and growth |
[79] |
|
3. |
3.1. Wnt signaling Decreased Wnt antagonist - WIF1 |
Increased LEF1 and β-catenin |
EMT induction and increased oral cancer progression |
[82,83] |
|
4. |
4.1. Matrix signaling Increased MMP1, MMP7, MMP9, and MMP2 and Collagen type 1 |
Elevated ERK/MEK level |
Degradation of the extracellular matrix and basement membrane |
[85] |
|
5. |
5.1. Notch signaling |
Increased Snail and reduced expression of E-cadherin |
Increased tumor metastasis |
[87,88] |
|
6. |
6.1. Hedgehog signaling SHH enhancement |
- |
Increased tumorigenesis via EMT induction |
[89,90] |
|
7. |
7.1. Phosphatase and tensin homolog deleted on chromosome 10 (PTEN) |
Decreased PTEN |
Cancer invasion via EMT induction |
[6] |
2- I also recommend including one figure that show the start and end of the signaling pathway,
Response 2: We have included the figure as per the comment.
Figure 1: Figure 1 shows the signaling events involved in the EMT during oral cancer via different receptors.
3- Some pathways are missing such as ELMO1 and Smad
Response: We have satisfied the given comment.
9.6. Engulfment and cell motility (ELMO) proteins and EMT in oral cancer
The impairment of TGFβ receptor II (Tgfbr2) causes cancer cell metastasis and in-vasion via ELMO1 repression. This ELMO1 is a guanine exchange factor via RAC. This ELMO1 acts as a proper target for the TGFβ and it restores the activities of the Tgfbr2. The impairment of the ELMO1 decreases the metastasis of the cancer cells [95]
3.1. Role of Transforming Growth Factor β (TGF-β) in oral cancer
TGF-β and its associated pathways play a twin role in the variation of its characteristics at its cellular level is called TGF-β Paradox [10]. TGF-β pathway is one of the most important pathways which mediate EMT. TGF-β plays an important role in the apoptosis of cancer cells and suppression of tumors in an environment with inflammation. In contrast, TGF-β enhances EMT by encouraging the migration of cancer cells via both smad signaling and non-smad signaling pathways [11]. This smad and non-smad signaling pathway is initiated by the ligands of the superfamily of TGF-β inclusive of TGF-β (3 isoforms) and bone morphogenic protein (BMP-2 to BMP-7; 6 isoforms) [12]. BMP-2 and BMP-7 are highly related to tumor differentiation and lymph node metastasis during oral cancer, showing a deprived diagnosis [13]. Impairing smad signaling of TGF-β leads to defeating the TGF-β inhibition effects and its proliferation. Either impairment of TGF- β or its attenuation affects its function involved in regulation since it is involved in the progression of tumor and carcinogenesis. Treatment of TGF-β in OSCC cells manifests a characteristic alteration of EMT by converting the cells like fibroblasts with E-cadherin attenuation and vimentin augmentation [14]. In OSCC, an induction of THBS-2 via TGF-β encourages cancer migration and increases the level of MMPs. It means it favors the invasion of OSCC [13]. The double function of TGF- β is highly found in oral cancer metastasis and is associated with its progression [15]. Superfamily member of TGF-β, such as BMP-2, impairs the level of Snail and N-cadherin; attenuates the level of CK9, further showing that BMP-2 is more involved in MET than EMT [16].
4-In the introduction, please include the prevalence of oral cancer, rate of death associated with oral cancer, and risk factors.
Response 4: We have added the data in the main manuscript and satisfied the comments.
Oral cancer is one of the most important health problems and is considered the main reason for the deaths of oral diseases. Oral squamous cell carcinoma (OSCC) is a large number of common malignancies, i.e., 90% cancer affected people belong to it from all the forms of oral cancer. This cancer development is mostly caused by increased abuse of alcohol, tobacco use, smoking, and HPV infection [1, 2]. According to the statistics globally for cancer in 2018, there is 2% of cases newly and 1.9% of cancer death. Approximately, 90% are OSCC in oral cancer. Three in one patient from oral cancer leads to recurrent and untreatable. This is mainly due to neoplasm recurrence, metastasis of cancer, and resistance to the drug which further causes less survival rate [3].
We thank the reviewers for their valuable suggestions to highlight the significance of this review and improve the review article to meet the journal’s standards. We hope that this revised manuscript is now better suited for publication in your esteemed journal.

Round 2
Reviewer 1 Report
The authors have adequately addressed most of the comments. However, the entire manuscript should be thoroughly checked for syntax and grammar errors before publication.
Author Response
The authors have adequately addressed most of the comments. However, the entire manuscript should be thoroughly checked for syntax and grammar errors before publication.
Response: Authors are thankful to the reviewer for evaluating the quality of the manuscript. We have thoroughly checked the entire manuscript and corrected all the grammatical errors. The track changed in the main manuscript.

Reviewer 3 Report
No further comments
Author Response
No further comments.
Response: All the authors are thankful to the reviewer for evaluating the quality of the manuscript.